# Familiar Dog or Familiar Person: Who Do Pet Dogs Best Synchronize with?

**DOI:** 10.3390/ani15040505

**Published:** 2025-02-11

**Authors:** Angélique Lamontagne, Thierry Legou, Marine Brunel, Thierry Bedossa, Florence Gaunet

**Affiliations:** 1Centre de Recherche en Psychologie et Neurosciences (UMR 7077), Faculté des Sciences site Saint Charles, Aix-Marseille University, Centre National de la Recherche Scientifique, 13331 Marseille, France; 2Association Agir pour la Vie Animale (AVA), 76220 Cuy-Saint-Fiacre, France; 3Laboratoire Parole et Langage (UMR 7309), Aix-Marseille University, Centre National de la Recherche Scientifique, 13100 Aix-en-Provence, France; 4Faculté des Sciences et Techniques, Université Jean Monnet, 42100 Saint-Etienne, France

**Keywords:** behaviour, canine, evolutionary convergence, interspecific interactions, synchronization

## Abstract

This study presents the first direct comparison of dog behavioural synchronization with a familiar human and a familiar dog, as previous research has examined each separately. This comparison offers a better understanding of canine cognition and how co-evolution with humans has affected dogs’ behaviours. We expected that, in an outdoor familiar setting, dogs would synchronize their behaviour more effectively with the familiar dog than with the familiar human, since shared locomotor patterns between members of the same species should facilitate behavioural alignment. The results confirmed a closer proximity when dogs were with the familiar dog than the familiar human. Speed adjustment was also greater when with the familiar dog, but this was observed only in dogs that did not live with other dogs. There were no significant differences in body alignment and gaze behaviour between the two experimental conditions. These findings demonstrate dogs’ ability to synchronize their behaviour with both other dogs and humans, with notable differences depending on whether they are interacting with an individual of the same or a different species. The study also highlights the impact of daily experiences, here cohabiting with other dogs, on behavioural synchronization in dogs.

## 1. Introduction

Dogs, as social animals, are sensitive to the behaviours of their conspecifics and use this social information to communicate and adjust their own behaviours [1]. Dogs are thus capable of synchronizing their behaviours with one another [2], and this intraspecific synchronization depends on familiarity [3]. Dogs synchronize more with familiar conspecifics than with unfamiliar ones, for instance, as evidenced by the closer proximity between familiar dogs during off-leash walks [4]. Dogs have inherited their capacity for behavioural synchronization from their ancestors, wolves [5], which are known for their cooperative abilities in group activities such as breeding and hunting [5,6,7,8]. The ability to synchronize behaviour with that of others is a key element in the social development of individuals and holds high adaptative value, as it is associated with increased survival, reduced energy expenditure, and greater social cohesion [9,10,11].

Contrary to wolves, dogs have the distinction of being a domesticated species. They were selected to assist and cooperate with humans in various tasks, such as guarding livestock and hunting. This led to the selection of specific traits in dogs, making them an intriguing subject in the investigation of the evolution of social cognition [12]. Research over the past decades has revealed that dogs are sensitive not only to the behaviours of their conspecifics, but also to those of humans [5,11,12,13]. Dogs exhibit social cognitive abilities towards humans that are typically seen at the intraspecific level, such as following ostensive and referential communication signals and synchronizing their behaviour with that of humans [14,15]. In recent years, there has been growing interest in the study of behavioural synchronization between dogs and humans [16,17,18,19,20,21,22,23]. These studies have shown that, when walking with humans, dogs stay close by, adjust their speed to match that of humans, and align their body orientation with that of humans. These abilities emerge early on in dogs, within the first few months of their life [24,25], depend on their level of familiarity with humans [21,26,27], and are similar to those found in humans [28].

To the best of our knowledge, no studies have directly compared dogs’ behavioural synchronization with conspecifics and with humans. The only comparative study available focuses on synchronization between dogs and humans versus between wolves and humans [19]. The authors found that domestic dogs synchronize more with humans than socialized wolves or pack-living dogs. This aligns with previous studies showing that dogs pay more attention to human behaviours than wolves do [29], and that wolves require extensive socialization to develop a comparable level of sensitivity to human actions [30]. Several studies have explored dogs’ sensitivity to conspecifics versus humans in contexts other than synchronized behaviours. Studies have reported that dogs are equally capable of learning socially from a conspecific and from a human during a problem-solving task [31] or in a local enhancement task [32]. Other research has demonstrated that dogs living in the same household display stronger affiliative behaviours towards each other than towards their human caregiver [3]. However, dogs tend to show greater visual attention towards humans than towards conspecifics [3,33]. Additionally, research has indicated that dogs behave differently when presented with a food reward in the presence of a conspecific or a human, exhibiting more avoidance behaviours (staying at a distance or looking away, for example) when a conspecific is present than when a human is present [34].

Taken together, these findings suggest that, while dogs possess similar social cognitive abilities towards humans and conspecifics [3,29], their responses to intraspecific and interspecific cues may vary depending on the task or ability being tested. Dogs appear to be, respectively, more sensitive to conspecifics with regard to affiliative behaviours and more attuned to humans in terms of referential communication, for instance. As intraspecific and interspecific synchronization have only been studied independently, however, it is difficult to draw conclusions about behavioural synchronization. This underscores the need for a direct comparison, which could provide valuable insights into dog cognition and the extent to which humans’ enculturation has impacted dogs’ behaviours.

In this study, we address this gap by comparing the behavioural synchronization of 20 dogs under two experimental conditions, with a familiar conspecific and with a familiar human, while spontaneously moving around an enclosed outdoor space. We ensured that the dogs’ level of familiarity with both the conspecific and the human was equal, as behavioural synchronization is known to depend on the level of familiarity, and we aimed to prevent its effects from affecting our results. Based on the definition provided by [9], we measured two components of behavioural synchronization: location synchronization (being in the same place at the same time) and activity synchronization (engaging in the same activity simultaneously). We hypothesized that it is easier to adjust to individuals with similar locomotor characteristics, so dogs would synchronize more with their conspecific than with the human. Specifically, we expected dogs to maintain a shorter distance (location synchronization) and a smaller difference in speed (activity synchronization), as well as spending more time with aligned body orientation (activity synchronization) when with their conspecific compared to the human. Additionally, we measured dogs’ gaze behaviour and hypothesized that dogs would exhibit greater visual attention towards the familiar human, thus spending more time looking at the familiar human than at the familiar conspecific, based on previous studies showing greater attention towards humans than towards conspecifics [3,33].

## 2. Materials and Method

### 2.1. Ethical Note

This study was conducted in accordance with the Declaration of Helsinki of 1975, which was revised in 2013, the French regulations for animal welfare (Rural Code Article R214-17 and the official French Legal Code of Animals, 2018), the institutional guidelines of Aix-Marseille University, and the internal regulations of the dog club Agility du Gier, in Saint-Joseph, France, where the study was carried out. The study was exempt from ethical approval because it was conducted with privately owned pet dogs without any physical constraints or invasive samplings. All dog owners were recruited on a voluntary basis and signed an informed consent form. All the dogs were in good health, with no signs of aging or behavioural problems, based on information from the owners and observations by the experimenters. Data collection took place in March and April 2024 at the canine club, a familiar environment for the dogs that regularly attended sessions there. Each dog was given 5 to 10 min prior to the observations to get acquainted with the GPS equipment and was free to move around the testing area throughout the entire experimental procedure. The dog’s owner always remained within the dog’s sight, and we visually controlled for stress-associated behaviours. During the experiment, none of the dogs exhibited any stress or fear signals.

### 2.2. Participants

The study included 20 adult dogs (10 males and 10 females) of various breeds or crossbreeds, aged between 1 and 8 years old (mean ± SD = 3.6 ± 2.3) and weighing a minimum of 7 kg (detailed information on the dogs’ breeds and other demographic data is available in the dataset deposited in the OSF repository). Of the 20 dogs, 12 were the sole dog in their household, while 8 lived with one or more other dogs (6 with one other dog and 2 with three other dogs). All the dogs regularly attended classes such as education, dog dancing, or agility at the canine club Agility du Gier once a week to once a month. They were accustomed to walking off leash and wearing a harness. Each dog was observed in two conditions, in a randomized order: with the familiar dog (with FD), a 7-year-old female Australian Shepherd, a member of the dog club, and with the familiar person (with FP), a 48-year-old male dog instructor at the dog club. Dogs’ behavioural synchronization with humans is not affected by their level of obedience training [20], but we verified that no verbal commands, instructions, or food rewards were provided during the observations to ensure that the dogs’ behaviours towards the FP were not influenced by training expectations.

The FD and the FP, who live in the same household, had been members of the club before the dogs in our study joined it, and they, thus, had known the dogs since they were registered at the club (mean duration ± SD = 3 ± 2.2 years at the time of the experiment). The FD and FP go to the club together once a week, so the dogs interact with them at the same frequency, ranging from one to four times per month (mean ± SD = 2.5 ± 1.4 times per month) during canine recess or walks outside of classes. The dogs did not attend the FP’s classes, but those of a different dog instructor. None of the dogs had ever exhibited fear or aggressive behaviours towards the FD or FP.

### 2.3. Experimental Set-Up and Apparatus

The dogs were observed individually in a 25 m × 25 m (625 m^2^) enclosed park at the dog club (Figure 1A). The owner and an experimenter (E1—the coauthor M.B.) were positioned in the centre of the test area during the observations. For half of the dogs, E1 was on the owner’s right (as illustrated in Figure 1), and, for the other half, E1 was on the owner’s left. Another experimenter (E2—the coauthor A.L.) remained outside the test area to monitor the experimental procedure. The owner and E1, who lived in the same household as the FD and FP, needed to be present inside the test area, as pre-tests without their presence showed that dogs and the FD stayed near the exit door of the test area and that dogs interacted poorly with the FD and FP. This behaviour aligned with previous findings indicating that dogs exhibit more motionlessness and reduced interactions with other individuals in the absence of their owner compared to when their owner is present [35,36,37].

The dogs, the FD, and the FP wore harnesses to which a Global Positioning System (GPS) rover was attached (Figure 1B). The GPS rovers were associated with a Real-Time Kinematic (RTK) base station which provided real-time corrections to the GPS rovers, with centimetre-level precision in positional data (see [20] for more details on the GPS rovers and RTK technology). Each rover recorded time, position, and the quality of the RTK correction every 0.25 s. The processing of GPS data resulted in four data points per second for the distance between the GPS rovers, and four data points per second for the speed of each rover. Two cameras fixed on tripods filmed the test area from opposite angles, and a GoPro© camera (designed by GoPro, Inc., based in San Mateo, CA, USA) was attached to the harness worn by the dogs (Figure 1B). E2 was positioned next to Camera 2 during the observations.

### 2.4. Experimental Procedure

When the owner and their dog arrived at the test site, E1 assisted the owner in fitting the harness equipped with the GPS rover and the GoPro© camera onto the dog’s back. E1 activated the GPS rover and the GoPro© camera, while E2 turned on Camera 1 and Camera 2 outside the test area. E1 and the owner stood still side-by-side in the centre of the park while the dog freely explored the area and became accustomed to the equipment for 5 to 10 min (Figure 2). The owner could communicate with their dog during the exploration phase. The FD and FP were in a closed building during this time so the dog could not see them and they could not see the dog. E1 and the owner stopped looking at and talking to the dog at the end of the exploration phase, and E2 brought the first agent (either the FD or the FP) to the entrance of the park, near Camera 2, while the second agent stayed in the building. E2 activated the GPS rover on the agent’s back and opened the gate to let the agent into the park. E2 stayed still and silent outside the testing area, and monitored the timing of the observation through the Camera 2 feed. The dog and the agent greeted each other, making eye contact and physical contact. The greeting phase was considered complete once the initial physical and eye contact ceased. The greeting phase with the first agent lasted (mean ± SD) 19.49 ± 8.48 s (with FD (*n* = 10) = 19.26 ± 10.50 s, with FP (*n* = 10) = 19.72 ± 6.41 s), based on visual inspection by E2 through Camera 2 and confirmed by video recordings checked by E1 and E2. The 5 min observation phase began immediately after the greeting phase. We chose this duration based on pre-test results, which indicated that it was sufficient to capture the dogs’ behaviours. E1, E2, and the owner did not move during this time. E2 signaled to E1 and the owner to leave the test area with the dog and the agent at the end of the 5 min. The cameras and GPS rovers were turned off, and a 15 min break followed, during which E1, E2, and the agent returned to the building, while the owner and their dog went for a short walk within the dog club premises. The dog did not wear the harness during the break. Observations were conducted on days without dog classes to ensure that no club members were present, and, as the dog club was a private location, no other individuals besides the dog, the owner, E1, E2, FD, and FP were present at the club, even during breaks. After the break, E1, E2, the owner, and the dog went back to the test area. The owner equipped the dog with the harness again, and E1 activated the GPS rover and the GoPro© camera, while E2 turned on Camera 1 and Camera 2 outside the test area. E1 and the owner positioned themselves back in the centre of the park and the dog was given 5 to 10 min to explore the area. E2 then brought the second agent to the entrance of the test area (Figure 2), activated the GPS rover on the agent’s back, and opened the gate to let the agent into the test area. The dog and the second agent greeted each other, the greeting phase lasting (mean ± SD) 20.38 ± 9.03 s (with FD (*n* = 10) = 23.61 ± 11.49 s, with FP (*n* = 10) = 17.14 ± 6.89 s), based on visual inspection by E2 through Camera 2 and confirmed by video recordings checked by E1 and E2. The 5 min observation phase with the second agent started after the greeting phase. E2 monitored the timing and indicated to E1 and the owner to leave the test area with the dog and the agent at the end of the 5 min observation phase. The dog remained off-leash, and no instructions were given to the dog during the entire experimental procedure. The whole procedure lasted approximately one hour. In some cases, several dogs were tested on the same day, but we do not believe that a previous test affected subsequent tests with other dogs due to the general abundance of canine scents at the dog club and the familiarity of all participating dogs with these scents.

### 2.5. Instructions for the Familiar Dog in the with FD Condition and for the Familiar Person in the with FP Condition

The FD received no instructions in the with FD condition and was allowed to move freely in the testing area when with the dogs. The FP was unaware of the study’s aim and hypotheses, but was given the following instructions prior to the experiment for the with FP condition, so that the observations of the dogs with the FP would be comparable to the observations of the dogs with the FD: “After entering the park, let the dog sniff you, extend your hand if you wish, then start walking whenever you like and keep your arms by your sides. Do not play with the dog or talk to the dog, but you can look at the dog and in the direction of E1 and the owner. Explore the entire area, including where E1 and the owner are, moving at one of four different paces: standing still, walking, slowly jogging, or running, spending more time walking and standing still than jogging, and running very little”. These instructions were derived from pre-tests conducted prior to the experiment, involving the observation of the FD with 6 dogs not part of the current study. Each dog was observed individually for 5 min with the FD in the park shown in Figure 1A. These pre-tests were video recorded, and the FD’s behaviours (see Appendix A) were coded by E1 and E2 using BORIS software (version 8.25.4, https://www.boris.unito.it/, accessed on 5 February 2025). The FP was not present during these pre-tests; E1 and each dog’s owner were present inside the test area, and E2 was outside the test area. The dogs and FD wore harnesses but were not equipped with a GPS.

### 2.6. Data Analysis

We analyzed data from the 5 min observation phases in R (version 4.4.1; https://www.R-project.org, accessed on 5 February 2025). We did not include the greeting phases in the analyses because they were solely intended for the dog and the agents to greet each other and were not relevant to the study, as individuals were close together and engaged in the same actions during these phases.

We first checked the locomotion and gaze behaviour of the FD and FP during the 5 min observations using Linear Mixed Models (LMMs). To model agents’ speed, we used the *glmmTMB* package, with a Gaussian error distribution, and we included the type of agent (2 levels: FD and FP) as fixed effects. Since the position data were recorded every 0.25 s, there was high temporal dependence, so we included an autocorrelation structure of order 1 (AR1). We modeled agents’ gaze behaviour towards the dogs using the *lme4* package, with the type of agent as a fixed effect. In both models, dog ID was included as a random effect.

To compare intra- and interspecific behavioural synchronization, we analyzed the following variables: the distance between the dog and the agent (location synchronization), the difference in speed between the dog and the agent (activity synchronization), and body alignment (another measure of activity synchronization). We also measured the dog’s gaze behaviour towards the agents; see the definitions of the variables in Table 1. A naïve coder who did not know the purpose and hypotheses of the study coded body alignment and gaze behaviour variables from the video recordings using BORIS software. The coder was first trained on videos from a different experiment until their results matched those previously obtained. To assess inter-coder reliability, a second naïve coder analyzed a subset of 25% of the videos, yielding a strong correlation between the two coders (time the agent spent gazing at the dogs: *R*^2^ = 0.98, *p* < 0.001; time the dogs spent gazing at the agent: *R*^2^ = 0.99, *p* < 0.001; body alignment: *R*^2^ = 0.97, *p* < 0.001).

We ran a Generalized Linear Mixed Model (GLMM) using the *glmmTMB* package to analyze the distance variable, with a gamma distribution for errors and log link function. The speed difference was modeled with a GLMM using the *glmmTMB* package with a ziGamma distribution for errors and log link function. We included an AR1 structure in these GLMMs. Body alignment and gaze behaviour were modeled with an LMM using the *lme4* package. In all (G)LMMs, the type of agent present (2 levels: with FD and with FP) and the presence of another dog in the household (2 levels: yes and no) as well as the interaction between these two factors were included as fixed effects. We included dogs’ sex and age as nuisance variables, and dog ID as a random effect.

Model assumptions were graphically checked using the DHARMa package. For each model, we compared the full model with the corresponding null model (which included only the nuisance variables and random factors) using Likelihood Ratio Tests (LRTs), with the anova function and the chisq argument. If the null–full model comparison was significant, we reported the estimates and standard errors (SE) of the full model, obtained using the summary function. We also provided 95% confidence intervals (95% CI), obtained using the confint function. Test statistics from Wald chi-square tests were calculated using the Anova function from the car package with Type 3 sum of squares. Where post hoc comparisons were necessary, we used the emmeans package with the Kenward–Roger approximation and Tukey corrections for multiple testing.

## 3. Results

### 3.1. Locomotion and Gaze Behaviour of the FD and FP

The FD and FP did not differ significantly in their speed (LRT: χ^2^ = 2.39, *df* = 1, *p* = 0.122) or in their gaze behaviour towards the dogs (LRT: χ^2^ = 1.04, *df* = 1, *p* = 0.308).

### 3.2. Location Synchronization of Dogs with the FD and FP

The null–full model comparison for the distance was significant (LRT: χ^2^  =  14.10, *df * =  3, *p*  =  0.003). The distance was smaller when the dogs were with the FD than when they were with the FP (Table 2 and Figure 3).

### 3.3. Activity Synchronization of Dogs with the FD and FP

The comparison between the null and full model for the speed difference was significant (LRT: χ^2^  =  9.57, *df * =  3, *p*  =  0.023). An analysis of the full model revealed a significant interaction between the type of agent present and the presence of another dog in the household (Table 3). Post hoc tests indicated that the speed difference was smaller for dogs not living with other dogs when they were with the FD than when they were with the FP (Table 4 and Figure 4).

Regarding body alignment, the null–full model comparison was not significant (LRT: χ^2^  =  2.52, *df * =  3, *p * =  0.472), indicating that the variables of interest did not significantly improve the model fit. The dog and the agent’s bodies were aligned for an average of 60.72 ± 26.44 s over the 5 min observation period.

### 3.4. Gaze Behaviour of Dogs Towards the FD and FP

The null–full model comparison was not significant (LRT: χ^2^  =  3.23, *df * =  3, *p*  =  0.356), which means that the variables we studied did not significantly improve the model fit. The dogs spent 136.36 ± 7.66 s gazing at the agent over the 5 min observation period.

## 4. Discussion

This study represents the first direct comparison of intraspecific and interspecific behavioural synchronization in dogs and provides key insights on the processes underlying behavioural synchronization. We compared dogs’ locomotor synchronization and gaze behaviour with a familiar conspecific (with FD) and with a familiar person (with FP) when spontaneously moving around in an outdoor setting. We hypothesized that dogs would synchronize more with the FD than with the FP, and that they would display greater visual attention towards the FP. Our hypotheses were partially confirmed: our findings indicate that dogs displayed better location synchronization and better speed adjustment with the FD, but this latter effect was found only for dogs living without another dog in their household. We found no significant differences between the two experimental conditions in terms of body alignment or direction of visual attention.

As expected, the closer distance between dogs and the FD, which aligns with our hypothesis, suggests a stronger proximity between conspecifics than between individuals from different species. Previous studies have reported lower proximity-seeking behaviours in shelter dog–caregiver dyads [38] and pet dog–child dyads [39] compared to dog–adult owner dyads. These differences were correlated with reduced synchronization levels in the two former dyads [21,26]. The enhanced location synchronization between dogs in our study may thus be attributed to stronger proximity-seeking behaviour between conspecifics. It is also possible that the congruency of locomotor patterns between conspecifics contributed to dogs maintaining a closer proximity to the familiar dog than to the familiar person. However, two studies have found that dogs seek proximity with an unfamiliar human more than with a cohabitant dog, except when the conspecific was their mother [40,41]. Future studies are thus needed to clarify links between proximity-seeking behaviours and behavioural synchronization in dogs.

Our findings on activity synchronization indicate that dogs adjust their speed more effectively to that of a familiar conspecific than to that of a familiar person, but this ability is influenced by cohabitation with another dog. A likely explanation for the stronger speed synchronization between dogs is physical: similar locomotor patterns make it easier for individuals to synchronize. In humans, locomotor synchronization occurs spontaneously during walks but is less easily achieved when individuals have different step frequencies or leg lengths [42]. Similarly, locomotor synchronization occurs spontaneously in dogs walking with humans, but the quadrupedal versus bipedal difference likely makes speed synchronization less easily achieved between dogs and humans than between conspecifics. Interestingly, this applies only for dogs that do not live in multi-dog households, as dogs living with conspecifics are better at synchronizing their speed with the human than dogs living without conspecifics. As dogs in multi-dog households are more visually exposed to interspecific interactions than dogs living without conspecifics, it is possible that this cohabitation enhances the dogs’ ability to adjust to human speed, due to repeated exposure to third-party interactions [43,44]. This finding aligns with the role of experience by observation in shaping behavioural synchronization, as noted in previous research [24]. It is proposed that behavioural synchronization develops when individuals perform an action while simultaneously observing others performing the same action, or when observing third parties performing synchronized actions [45,46]. Our study is in line with this view, highlighting how daily experience influences dogs’ behaviours towards humans and conspecifics.

Despite the closer proximity and better speed adjustment with the FD for some dogs, we did not find a significant difference in the body alignment of the dogs when with the FD and with the FP, which was contrary to our expectations. A possible explanation is that dogs have more experience of body alignment with humans than with conspecifics. Daily walks with humans may indeed foster the tendency to align their bodies with humans when walking in the same direction. However, further research is needed, as only one study has measured the body alignment between dogs and humans [21]. The authors found that dogs spent about 30% of the time aligned with a child’s body orientation during walks, and we found a 20% alignment rate in our study. To our knowledge, no other study has measured the body alignment between dogs. Due to the lack of significant results in our study, we cannot draw definitive conclusions about body alignment. Moreover, we did not find significant differences in the direction of dogs’ visual attention between the two experimental conditions. It is possible that the increased attention towards humans described in the previous literature [3] may only be exhibited in specific contexts, such as unsolvable tasks, and not in free interactions without a specific goal. Once again, we cannot extend our interpretation further on this point given the non-significant results.

This study contributes to the growing body of research comparing intraspecific and interspecific cognition in dogs. Our findings that dogs exhibited intraspecific behavioural synchronization, which surpassed some components of behavioural synchronization with a person, suggest that the ability to synchronize with conspecifics is inherent to the species. This study is a new contribution to the hypothesis that intraspecific behavioural synchronization in dogs and in humans may thus result from evolutionary convergence. Although dogs and humans are phylogenetically distant, shared environmental factors have certainly shaped this convergence [47]. This explains the striking similarity in social cognitive abilities between dogs and young children [14]. Moreover, dogs’ synchronization abilities extend beyond the intraspecific level to interspecific interactions with humans. The development of interspecific behavioural synchronization is likely the outcome of domestication and the ontogenetic experience of pet dogs in human environments [48]. This aligns with studies showing that canine cognitive skills towards humans, such as referential communication and focused attention (visual attention directed to a specific human’s body part such as eyes or gestures), emerged due to dogs’ history of domestication and their enculturation with humans [10,23,25]. One hypothesis suggests that these social cognitive skills were at first present within conspecifics and that, during domestication, they were subsequently transferred to, and reinforced by, interactions with humans [12]. This hypothesis is supported by the fact that wolves, the ancestors of dogs, exhibit cooperative behaviours among conspecifics [5]. The observed intraspecific and interspecific behavioural synchronization in our study is in line with this hypothesis. However, another hypothesis posits that humans, as an integral part of dogs’ social environment and as highly competent social partners, have influenced natural and artificial selection to foster novel forms of social abilities in dogs, specifically exhibited during interactions with humans [12]. Given the complexity of domestication, it is unlikely that any single mechanism fully explains the cognitive abilities observed in modern dogs. Furthermore, developmental and lifelong experiences also significantly shape dogs’ cognitive capacities towards humans [10,24,48]. Our study further highlights the influence of experience, as demonstrated by the finding that cohabitation with a conspecific affects speed synchronization in dogs, shedding light on the flexibility of behavioural synchronization.

Our study presents certain limitations. One potential bias stems from our sample population, which consisted of dogs accustomed to training and frequent interaction with conspecifics. Moreover, half of the dogs were herding breeds. Although we made this selection for methodological reasons, future studies should include other dog populations or dog breeds, which would help to generalize our findings. A similar study conducted with shelter dogs, for example, could examine the impact of social deprivation on these dogs’ ability to synchronize with humans and with conspecifics. Comparing different breeds of dogs could also be valuable for studying the effects of size, leg length, and breed group on dogs’ behaviour. Differences in leg length may indeed influence dogs’ locomotor synchronization, and breed groups can differ in their behaviour [49]: sheepdogs, for instance, have been selected to be particularly attuned to humans. Another possible limitation of our study was the presence of humans. The owner’s presence in the testing area was necessary in our study, as the absence of the owner in pre-tests significantly reduced the number of interactions between the dogs and the agents. Consequently, our protocol did not allow us to observe conspecific interactions without a human presence. Moreover, for methodological reasons, we conducted our study in a familiar environment, with a familiar dog and a familiar human. It could be interesting, in future research, to explore the impact of familiarity or unfamiliarity with the test area or with the interacting partner on the dogs’ behaviour. Furthermore, the participation of the same familiar dog and person for all dogs ensured comparable observations and effectively controlled for the effect of familiarity, an important social modulator of behavioural synchronization. However, the results could be different if other agents were used to play the roles of the familiar dog and human. Replicating the protocol with dogs from the same household and a human household member, for instance, could extend our results and shed light on how the level of familiarity impacts the comparison of intraspecific and interspecific behavioural synchronization.

## 5. Conclusions

This study marks the first comparison of dog behavioural synchronization at intraspecific and interspecific levels. Our findings demonstrate that dogs spontaneously synchronize with both a familiar conspecific and a familiar human. Moreover, dogs maintain a closer proximity to the conspecific, and cohabitation with another dog influences dogs’ speed synchronization. This research deepens our understanding of the evolution of behavioural synchronization in dogs. Prior research revealed that this social skill was present at an intraspecific level in dogs’ ancestors, and we have shown in this study that modern dogs continue to possess this skill. The capacity for synchronization thus represents an evolutionary convergence between canids and humans, and domestication and ontogeny likely fostered the further development of this ability in dogs during interactions with humans.

## Figures and Tables

**Figure 1 animals-15-00505-f001:**
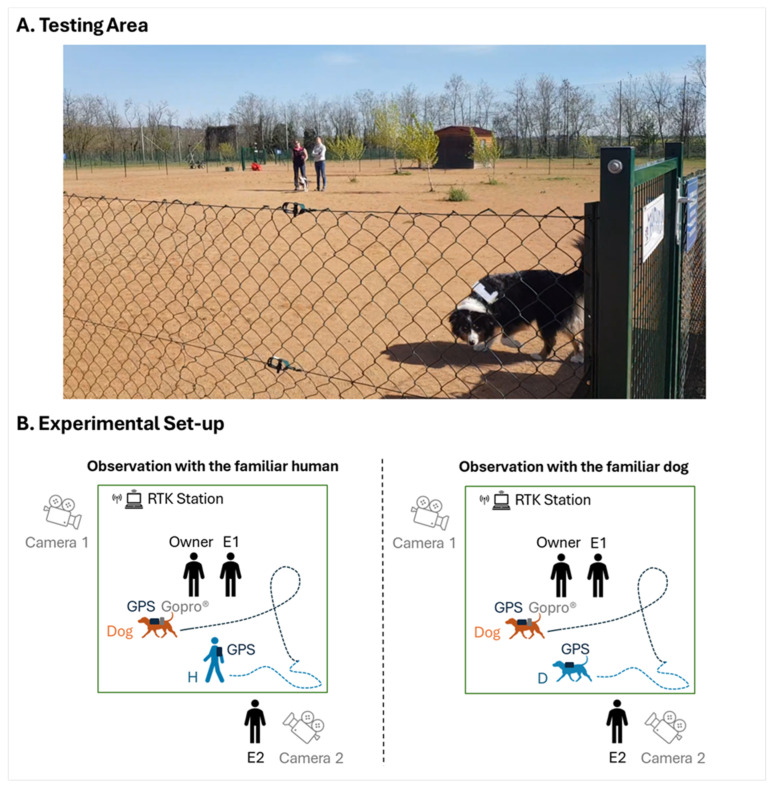
(**A**) Testing area. In this example, the owner is on E1’s right side, the FD is near the exit gate, and the dog is near E1 and the owner. (**B**) Experimental set-up for observations with the FD and with the FP.

**Figure 2 animals-15-00505-f002:**
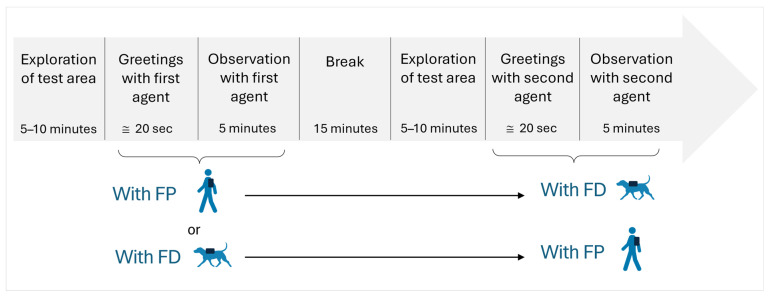
Chronology of experimental procedure: 10 dogs were first observed with the FP and then with the FD; the other 10 dogs were first observed with the FD and then with the FP.

**Figure 3 animals-15-00505-f003:**
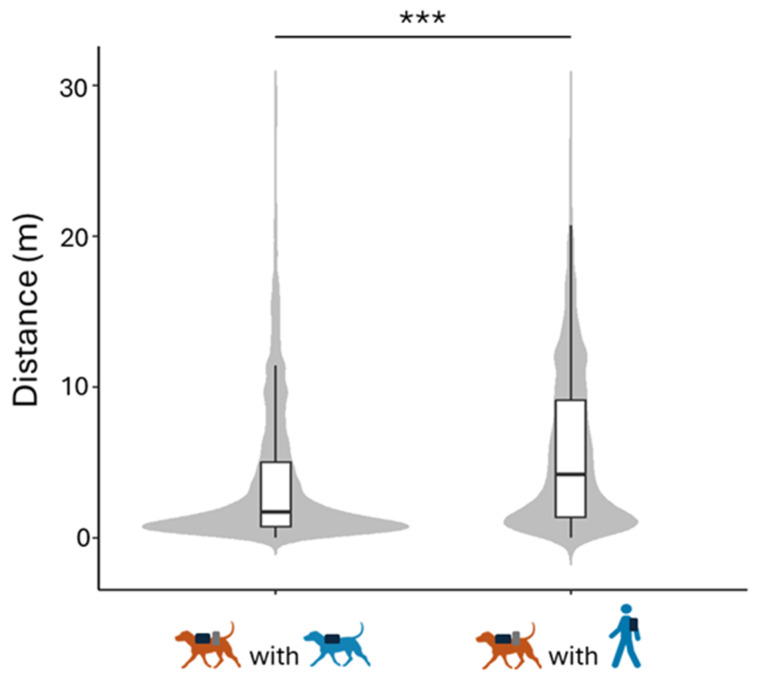
Effect of type of agent on location synchronization of dogs. With FD and with FP conditions are depicted in blue; the dogs are depicted in brown. The distance between the dogs and FD was significantly shorter than between the dogs and the FP. *** *p* ≤ 0.001.

**Figure 4 animals-15-00505-f004:**
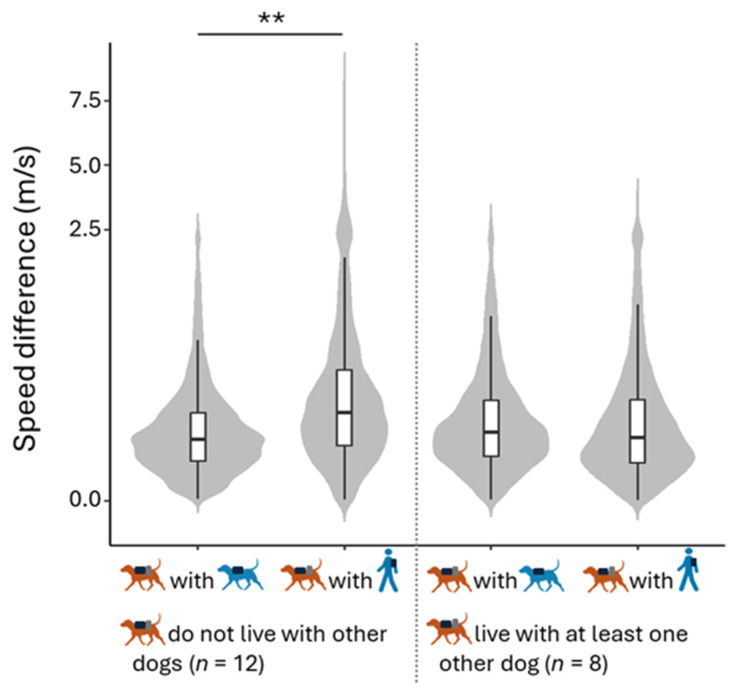
Effect of interaction between type of agent present and presence of another dog in the household on speed difference. With FD and with FP conditions are depicted in blue; the dogs are depicted in brown. For dogs that do not live with other dogs, the speed difference between the dogs and FD was significantly smaller than between the dogs and the FP. Data exceeding 2.5 m/s were squashed. ** 0.001 ≤ *p* ≤ 0.01.

**Table 1 animals-15-00505-t001:** Definitions of the dependent variables.

	Variable Name	Variable Definition	Data Collection
Location synchronization	Distance between agent and dog	Distance in metres between the agent’s back and the dog’s back.	From GPS data, 2400 values for each dog (1200 for each agent).
Activity synchronization	Speed difference between agent and dog	Absolute difference in metres/second between the agent’s speed and the dog’s speed.	From GPS data, 2400 values for each dog (1200 for each agent).
	Body alignment	Time in seconds during which the dog’s body and the agent’s body were oriented in the same direction.	From video recordings, 2 values for each dog (1 for each agent).
Gaze behaviour	Gaze towards the agent	Time in seconds during which the dog’s muzzle was oriented towards the agent.	From video recordings, 2 values for each dog (1 for each agent).

**Table 2 animals-15-00505-t002:** Results of the GLMM of the distance between the dog and the FD or FP. The predictors were dummy-coded, with type of agent present (with FD), presence of another dog in the household (no), and sex (female) as the reference categories. Estimate  ±  SE refers to the difference of the response between the reported level of this categorical predictor and the reference category of the same predictor.

Fixed Effect	Estimate	SE	*z* Value	95% CI	χ^2^	*p*
Intercept	0.80	0.25	3.25	[0.32; 1.28]	10.59	0.001
Type of agent present (with FP)	0.75	0.19	4.02	[0.38; 1.11]	16.19	<0.0001
Presence of another dog in the household (yes)	0.08	0.26	0.30	[−0.43; 0.58]	0.09	0.767
Type of agent present (with FP): Presence of another dog in the household (yes)	−0.40	0.29	−1.35	[−0.97; 0.18]	1.83	0.176
Dog’s age	0.04	0.04	0.96	[−0.04; 0.11]	0.93	0.336
Dog’s sex (male)	−0.19	0.18	−1.09	[−0.54; 0.15]	1.18	0.278

**Table 3 animals-15-00505-t003:** Results of the GLMM of the speed difference. The predictors were dummy-coded, with type of agent present (with FD), presence of another dog in the household (no), and sex (female) as the reference categories. Estimate  ±  SE refers to the difference of the response between the reported level of this categorical predictor and the reference category of the same predictor.

Fixed Effect	Estimate	SE	*z* Value	95% CI	χ^2^	*p*
Intercept	−0.85	0.16	−5.19	[−1.17; −0.53]	26.97	<0.0001
Type of agent present (with FP)	0.40	0.12	3.35	[0.16; 0.63]	11.19	<0.001
Presence of another dog in the household (yes)	0.16	0.14	1.21	[−0.10; 0.43]	1.45	0.228
Type of agent present (with FP): Presence of another dog in the household (yes)	−0.48	0.19	−2.57	[−0.85; −0.11]	6.58	0.010
Dog’s age	−0.01	0.03	−0.21	[−0.06; 0.05]	0.04	0.835
Dog’s sex (male)	0.06	0.13	0.43	[−0.21; 0.32]	0.19	0.666

**Table 4 animals-15-00505-t004:** Post hoc tests for the interaction between type of agent present (with FD or FP) and presence of another dog in the household (yes or no) on the speed difference variable.

Contrasts	Ratio	SE	95% CI	Z Ratio	*p*
With FD—no/with FP—no	0.67	0.08	[0.50; 0.91]	−3.35	0.005
With FD—no/with FD—yes	0.85	0.85	[0.60; 1.20]	−1.21	0.624
With FD—no/with FP—yes	0.92	0.92	[0.57; 1.49]	−0.43	0.973
With FP—no/with FD—yes	1.26	1.26	[0.82; 1.96]	1.37	0.519
With FP—no/with FP—yes	1.37	1.37	[0.80; 2.37]	1.50	0.438
With FD—yes/with FP—yes	1.09	1.09	[0.75; 1.58]	0.68	0.939

## Data Availability

The data presented in this study are openly available in the Open Science Framework repository at the following URL (accessed on 9 October 2024): https://osf.io/naxyp/?view_only=64d31c1f4ead417c958df9ebb6d26c86.

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
