# Peer review of "Familiar Dog or Familiar Person: Who Do Pet Dogs Best Synchronize with?"

_animals, 2025, doi:10.3390/ani15040505_

Round 1

Reviewer 1 Report

Comments and Suggestions for Authors

The manuscript with the topic "Familiar dog or familiar person: Who do pet dogs best synchronize with?" (Manuscript ID animals-3427913) is written in a professional and scientific language. 

I would like to suggest some thingя to be improved:

1.       You made your research in very well-known area in the dog’s club. You have in the title “Familiar dog” and ”Familiar person”, but no “Familiar area”. In my opinion you must make the same research in unknown area with the same animals and the same “Familiar person” and to compare the results.

2.       In my opinion is very important to make the research with certain breed/breeds and to compare the results with the present. 

REFERENCES

28.9% of the references are for a period of the last 5 years. It is mandatory to increase their number.

Author Response

The manuscript with the topic "Familiar dog or familiar person: Who do pet dogs best synchronize with?" (Manuscript ID animals-3427913) is written in a professional and scientific language. 

I would like to suggest some thingя to be improved:

Comment 1: You made your research in very well-known area in the dog’s club. You have in the title “Familiar dog” and ”Familiar person”, but no “Familiar area”. In my opinion you must make the same research in unknown area with the same animals and the same “Familiar person” and to compare the results.

Response 1: Thank you for this suggestion. We did not include “familiar area” in the title to keep it concise, but our study indeed focused on intraspecific and interspecific behavioural synchronization in a familiar environment. We have clarified this at the beginning of the abstract, line 21-23 “We expected that, in an outdoor familiar setting, dogs would synchronize their behaviour more effectively with the familiar dog than with the familiar human”. We chose a familiar area for methodological reasons.

Additionally, we agree that the impact of familiarity or unfamiliarity with the environment or with the human and dog could be explored in future research. However, this was not the objective of our study, which aimed to compare for the first time intraspecific and interspecific behavioural synchronization. We have included this suggestion for future research in the discussion section, line 464-466: “Also for methodological reasons, we conducted our study in a familiar environment, with a familiar dog and a familiar human. It could be interesting, in future research, to explore the impact of familiarity or unfamiliarity with the test area or with the interacting partner on the dogs’ behaviour.”

Comment 2: In my opinion is very important to make the research with certain breed/breeds and to compare the results with the present. 

Response 2: We agree that it would be interesting to compare the results for different breeds. We have added this suggestion at the end of the discussion section, line 452-460: “Although we made this selection for methodological reasons, future studies should include other dog populations or dog breeds, which would help to generalize our findings. A similar study conducted with shelter dogs, for example, could examine the impact of social deprivation on these dogs’ ability to synchronize with humans and with conspecifics. Comparing different breeds of dogs could also be valuable for studying the effects of size, leg length, and breed group on dogs’ behaviour. Differences in leg length may indeed influence dogs’ locomotor synchronization, and breed groups can differ in their behaviour[52]: sheepdogs, for instance, have been selected to be particularly attuned to humans.”

Comment 3: REFERENCES

28.9% of the references are for a period of the last 5 years. It is mandatory to increase their number.

Response 3: We have added more recent references (please see the references section), so the proportion of references from the last 5 years is now 44.2%.

Reviewer 2 Report

Comments and Suggestions for Authors

very interesting research experiment with a typical behavioral theme.

Research hypothesis formulated understandably.

Interestingly designed experiment with a familiar person and a familiar dog.

However, I have a few questions for the Authors:

Why didn't they use temperament tests in the dogs being assessed? After all, the reaction time and its intensity can be conditioned by the reactivity of the central nervous system.

Why wasn't stress behavior assessed during the experiment, e.g. licking or opening the mouth? Or wasn't a heart rate monitor used to assess HR during the 5 minutes of the experiment?
After all, stress largely determines behavior depending on the strength of the external environmental stimuli.
In my opinion, with such a large number of dogs, there should be no division into dogs kept individually and dogs kept with other dogs.

Why is there no information whether there are one or more household members in the household?

Why wasn't the so-called zero sample used - e.g. a strange dog and a strange person, as a reference to the results obtained?

Author Response

Comment 1: Why didn't they use temperament tests in the dogs being assessed? After all, the reaction time and its intensity can be conditioned by the reactivity of the central nervous system.

Response 1: Thank you for the nice general comment. We did not use temperament tests because our study focused on intraspecific vs. interspecific locomotor synchronization. It had not been investigated before, so our study was a first approach to this comparison. Moreover, we used a within-subject design, so the same dogs were observed in both experimental conditions. Therefore, we are not certain that a temperament test would have had a significant impact on this comparison. Nevertheless, it could be interesting for future research to examine the effect of temperament on locomotor synchronization in dogs.

Comment 2: Why wasn't stress behavior assessed during the experiment, e.g. licking or opening the mouth? Or wasn't a heart rate monitor used to assess HR during the 5 minutes of the experiment?
After all, stress largely determines behavior depending on the strength of the external environmental stimuli.

Response 2: We did not use a heart rate monitor during the experiment, but we monitored behavioural signs of stress through visual observation, as stated in the Ethical Note, line 144–145: “The dog’s owner always remained within the dog’s sight, and we visually controlled for stress-associated behaviours”. Also, we recruited dogs that were accustomed to interacting with the familiar dog and the familiar human in this experimental environment. Unsurprisingly, we did not detect any signs of stress or fear during our observations. We have clarified this point in the Ethical Note, line 145–146: “During the experiment, none of the dogs exhibited any stress or fear signals.

Comment 3: In my opinion, with such a large number of dogs, there should be no division into dogs kept individually and dogs kept with other dogs.

Response 3: We understand your perspective. Nevertheless, we anticipated that dogs might behave differently depending on whether they live with another dog or not, and we did observe differences based on this factor in our results. We thus believe that this distinction is important, as it provides insights into how dogs’ daily life experience impacts their behaviour.

Comment 4: Why is there no information whether there are one or more household members in the household?

Response 4: If you are referring to the number of dogs living in the same household as the dogs participating in the study, this information is available in the dataset provided, as stated line 151-152: "Detailed information on the dogs’ breeds and other demographic data is available in the dataset deposited in the OSF repository." If you are referring to human or non-dog household members, we acknowledge that this could have provided additional details, but we did not collect this information because we did not consider it directly relevant to our study. Indeed, we focused on dog-dog and dog-human interactions, and we knew that all participating dogs interacted with at least one human in their household and with or without at least one other dog.

Comment 5: Why wasn't the so-called zero sample used - e.g. a strange dog and a strange person, as a reference to the results obtained?

Response 5: We did not use a zero sample because our objective was not to compare our results with those obtained with unfamiliar partners, but rather to determine whether a dog synchronizes more with a familiar conspecific than with a familiar human. Comparing our results with a situation where the dog and human are unfamiliar, or in an unfamiliar environment, would indeed be very interesting and could be explored in future studies. We have added this perspective at the end of the discussion, line 463-467: “Also for methodological reasons, we conducted our study in a familiar environment, with a familiar dog and a familiar human. It could be interesting, in future research, to explore the impact of familiarity or unfamiliarity with the test area or with the interacting partner on the dogs’ behaviour.”

Reviewer 3 Report

Comments and Suggestions for Authors

Thank you for the opportunity to understand your work.  I appreciate the time, effort, and dedication that went into your work.  I do think you have an interesting paper and one that could lead to further exploration.  I suggest building on this work.

Admittedly, as I sit on the couch with my two dogs lying next to me, I think I had as many questions arise as where answered.  As background, my 2 year old Morkie and 3 year old mixed Breed are both "rescues" - one coming from an Amish community (she has a TBI) and one from an out-of-state animal shelter.  Neither are "formally" trained but they are socialized as to the expectations of the household - e.g. come when called, go potty outside, don't rush the door when it is opened, don't counter surf for food, food is served at anticipated times of the day, etc.  I mention this because I wonder how dogs that regularly "go" somewhere for formal instruction may differ from my two couch potatoes.  Some of this was noted in your report as challenges but I am concerned that the formal training of the dogs may have an unknown influence.  Replicating the study with non-trained dogs may be a way to tease this out.   I also will add that one of my dogs is 12 lbs and the other is 30 lbs with a large difference in height.  As both of my dogs have "terrier" in them, I do know that instinct supersedes my instruction.  If they see a squirrel, they simply will not stop in their tracks and come back when I call - so I think a more diverse selection of dogs based on perceived breed may needed as well.  Herding dogs are, by definition, dogs that share an instinctual ability to control the movement of other animals - so I can't help but wonder what role this may play in the FD or the FP scenarios.....

I am making these suggestions as consideration for additional or further areas to explore.  That is, clearly the sample is not generalizable nor representative - and you did address this in your Limitations - but it is a beginning to exploration that I think would make more significant contributions over time.  Likewise, others may be encouraged to replicate this work under different conditions with different breeds as well.

IAC, here are the notes I made for myself that I wish to share with you:

Notes on:   Familiar dog or familiar person: Who do pet dogs best synchronize with?

Please note that I will make notes as I read along (“real time”).  Grammatical concerns will be noted in bold.  Highlights reflect specific questions or concerns.

Line 48:  I am expecting that this will be elaborated upon but wanted to note for myself to watch for it….. what are the conditions for “when walking”?  e.g. leashed? How short/long is leash? Or untethered?  What are the environmental conditions? Neighborhood sidewalk? City or Local Park? City Street?

Additionally, how does breed and size difference between the dogs factor? For example, I have a mixed breed who resembles a Harrier and a Morkie and their gait are very different due to leg length.  Also, look for dog’s length of time with FP, if dog was with FP since puppyhood or adopted later in life, etc.   Lastly, some breeds may have behavioral traits that will impact study e.g. herding dogs vs hunting dogs vs mixed breeds

     Line 58:  Off leash walks were noted.  *All of the questions I had regarding Line 48 noted above have been answered.

Line 82:  Singular/plural issue. “… with a previous studies”….. 

The graphics really helped visualize! 

Admittedly, at first I was very concerned that the dogs studied here were ‘trained’ dogs in that they had not only a history with the FP but also of the location. As training represents learning through external social control (e.g. corrections), have previous behavior patterns been “trained out”?  That is, do dogs that have been trained have the same mechanism by which they exercise “choice”?

Line 372 and thereabouts:  I wonder if, in the case of a cohabitant dogs, that dogs who have already worked out their “place” relative to one another, if that played a role.  E.g. one dog defers to the other more often.

       Line 381:   Noted the differences in gait (e.g. leg length)

Line 429:  Consider comma offsets.  “…..they were subsequently transferred to and reinforced by interactions with humans” with a comma after the word “to” and after the word “by”

References:  I will note that there are some dates/years that are in bold when they should not be bolded (this may also be a change in font).  This needs to be double-checked.

Also double check other aspects of References.  For example, citation #32 may have an incomplete title of the Journal.  The Journal is Behaviour (not Behav).

Author Response

Comment 1:Thank you for the opportunity to understand your work.  I appreciate the time, effort, and dedication that went into your work.  I do think you have an interesting paper and one that could lead to further exploration.  I suggest building on this work.

Admittedly, as I sit on the couch with my two dogs lying next to me, I think I had as many questions arise as where answered.  As background, my 2 year old Morkie and 3 year old mixed Breed are both "rescues" - one coming from an Amish community (she has a TBI) and one from an out-of-state animal shelter.  Neither are "formally" trained but they are socialized as to the expectations of the household - e.g. come when called, go potty outside, don't rush the door when it is opened, don't counter surf for food, food is served at anticipated times of the day, etc.  I mention this because I wonder how dogs that regularly "go" somewhere for formal instruction may differ from my two couch potatoes.  Some of this was noted in your report as challenges but I am concerned that the formal training of the dogs may have an unknown influence.  Replicating the study with non-trained dogs may be a way to tease this out.   I also will add that one of my dogs is 12 lbs and the other is 30 lbs with a large difference in height.  As both of my dogs have "terrier" in them, I do know that instinct supersedes my instruction.  If they see a squirrel, they simply will not stop in their tracks and come back when I call - so I think a more diverse selection of dogs based on perceived breed may needed as well.  Herding dogs are, by definition, dogs that share an instinctual ability to control the movement of other animals - so I can't help but wonder what role this may play in the FD or the FP scenarios.....

I am making these suggestions as consideration for additional or further areas to explore.  That is, clearly the sample is not generalizable nor representative - and you did address this in your Limitations - but it is a beginning to exploration that I think would make more significant contributions over time.  Likewise, others may be encouraged to replicate this work under different conditions with different breeds as well.

IAC, here are the notes I made for myself that I wish to share with you:

Notes on:   Familiar dog or familiar person: Who do pet dogs best synchronize with?

Please note that I will make notes as I read along (“real time”).  Grammatical concerns will be noted in bold.  Highlights reflect specific questions or concerns.

Response 1: We truly appreciate your positive and encouraging feedback on our work. As you have pointed out, this is a first study on the topic, and we fully recognize that more research will be necessary to generalize our findings. We agree that it would be valuable in the future to explore factors such as dog breeds, education levels, and familiarity with the study area. We have specifically addressed your points regarding the size/weight of the dogs and their level of training in our responses to your comments below. Your comments have given us valuable insights, and we have really enjoyed reflecting on them.

Comment 2: Line 48:  I am expecting that this will be elaborated upon but wanted to note for myself to watch for it….. what are the conditions for “when walking”?  e.g. leashed? How short/long is leash? Or untethered?  What are the environmental conditions? Neighborhood sidewalk? City or Local Park? City Street?

Response 2: As you noted below, the answers to these questions are provided a few lines further down in the manuscript.

Comment 3: Additionally, how does breed and size difference between the dogs factor? For example, I have a mixed breed who resembles a Harrier and a Morkie and their gait are very different due to leg length.  Also, look for dog’s length of time with FP, if dog was with FP since puppyhood or adopted later in life, etc.   Lastly, some breeds may have behavioral traits that will impact study e.g. herding dogs vs hunting dogs vs mixed breeds

Response 3: We did not include breed or size as factors in our analysis, because our sample consisted largely of herding dogs along with a diverse range of other breeds. The participating dogs weighed between 7 and 35 kg, we agree that variations in size and weight can influence locomotion, so future studies should further explore this aspect by specifically testing for size effects or comparing different breeds. We have added this perspective to the discussion, line 456-460: “Comparing different breeds of dogs could also be valuable for studying the effects of size, leg length, and breed group on dogs’ behaviour. Differences in leg length may indeed influence dogs’ locomotor synchronization, and breed groups can differ in their behaviour[52]: sheepdogs, for instance, have been selected to be particularly attuned to humans.”

Regarding the length of time dogs have known FP, this information is available in the dataset deposited in the repository. As FP and FD always go the club together, all the dogs have known FP and FD for the same length of time (ranging from 1 to 6 years, with an average of 3 years). We acknowledge that there is some variability in this duration, but we do not believe that this affects our synchronization comparison, as we used a within-subject design and each dog has known FP and FD from the same time and interacts with each of them at the same frequency.

     Line 58:  Off leash walks were noted.  *All of the questions I had regarding Line 48 noted above have been answered.

Comment 4: Line 82:  Singular/plural issue. “… with a previous studies”….. 

Response 4: We have corrected the singular/plural issue by removing "a”, thank you.

Comment 5: The graphics really helped visualize! 

Response 5: Thanks for the nice comment!

Comment 6: Admittedly, at first I was very concerned that the dogs studied here were ‘trained’ dogs in that they had not only a history with the FP but also of the location. As training represents learning through external social control (e.g. corrections), have previous behavior patterns been “trained out”?  That is, do dogs that have been trained have the same mechanism by which they exercise “choice”?

Response 6: Thank you for raising this interesting point. It is true that the dogs were used to be trained, which means that they have a higher level of education than average. Some of them know how to walk to heel, for example. This level of education could potentially influence their behaviour in general. However, even though the experiment took place at their training site, they also have canine recreation at the dog club. Indeed, after training sessions, the dogs have the opportunity to unwind and express their spontaneous behaviours without any commands or expectations. They therefore have both a “work mode” and a “recreation mode”. In the context of our experiment, we ensured that the dogs were not in “work mode”: they were not waiting for commands or rewards. We are not certain whether their “work mode” had any influence on their “recreation mode”, but we did not observe any behaviours that differed from what would typically be seen, during a regular walk for example. That said, it would be interesting to explore this further by replicating the experiment with untrained dogs.

Comment 7: Line 372 and thereabouts:  I wonder if, in the case of a cohabitant dogs, that dogs who have already worked out their “place” relative to one another, if that played a role.  E.g. one dog defers to the other more often.

Response 7: In the case of dogs living in the same household, we agree that their respective roles during daily interactions probably influence their behaviour. These roles can vary greatly: some dogs adopt a more deferential position most of the time, and for others, the roles shift depending on the context, the type of interaction, etc. This aspect would be interesting to explore in future studies.

Comment 8: Line 381:   Noted the differences in gait (e.g. leg length)

Response 8: Yes, gait differences affect synchronization in humans.

Comment 9: Line 429:  Consider comma offsets.  “…..they were subsequently transferred to and reinforced by interactions with humans” with a comma after the word “to” and after the word “by”

Response 9: Done, thank you (line 437).

Comment 10 : References:  I will note that there are some dates/years that are in bold when they should not be bolded (this may also be a change in font).  This needs to be double-checked.

Also double check other aspects of References.  For example, citation #32 may have an incomplete title of the Journal.  The Journal is Behaviour (not Behav).

Response 10: Thank you for pointing this out. We have carefully reviewed the reference formatting to comply with the journal’s requirements.